# Differential Positioning with Bluetooth Low Energy (BLE) Beacons for UAS Indoor Operations: Analysis and Results

**DOI:** 10.3390/s24227170

**Published:** 2024-11-08

**Authors:** Salvatore Ponte, Gennaro Ariante, Alberto Greco, Giuseppe Del Core

**Affiliations:** 1Department of Engineering, University of Campania “L. Vanvitelli”, 81031 Aversa, Italy; salvatore.ponte@unicampania.it; 2Department of Science and Technology, University of Naples “Parthenope”, 80133 Naples, Italy; alberto.greco@uniparthenope.it (A.G.); giuseppe.delcore@uniparthenope.it (G.D.C.)

**Keywords:** UAS, indoor positioning system, BLE beacon, *RSSI*, trilateration, extended Kalman filter, differential distance correction

## Abstract

Localization of unmanned aircraft systems (UASs) in indoor scenarios and GNSS-denied environments is a difficult problem, particularly in dynamic scenarios where traditional on-board equipment (such as LiDAR, radar, sonar, camera) may fail. In the framework of autonomous UAS missions, precise feedback on real-time aircraft position is very important, and several technologies alternative to GNSS-based approaches for UAS positioning in indoor navigation have been recently explored. In this paper, we propose a low-cost IPS for UAVs, based on Bluetooth low energy (BLE) beacons, which exploits the *RSSI* (received signal strength indicator) for distance estimation and positioning. Distance information from measured *RSSI* values can be degraded by multipath, reflection, and fading that cause unpredictable variability of the *RSSI* and may lead to poor-quality measurements. To enhance the accuracy of the position estimation, this work applies a differential distance correction (DDC) technique, similar to differential GNSS (DGNSS) and real-time kinematic (RTK) positioning. The method uses differential information from a reference station positioned at known coordinates to correct the position of the rover station. A mathematical model was established to analyze the relation between the *RSSI* and the distance from Bluetooth devices (Eddystone BLE beacons) placed in the indoor operation field. The master reference station was a Raspberry Pi 4 model B, and the rover (unknown target) was an Arduino Nano 33 BLE microcontroller, which was mounted on-board a UAV. Position estimation was achieved by trilateration, and the extended Kalman filter (EKF) was applied, considering the nonlinear propriety of beacon signals to correct data from noise, drift, and bias errors. Experimental results and system performance analysis show the feasibility of this methodology, as well as the reduction of position uncertainty obtained by the DCC technique.

## 1. Introduction

The possibility of gathering real-time information from the environment by means of sensors onboard smart devices and objects has paved the way for the concept of Internet of Things (IoT), with which all smart things, such as devices, sensors, and industrial and utility components, are interconnected through networks, changing our way of living, working, and studying [1]. IoT is becoming important for context awareness [2], network management and security, health monitoring [3], personal delivery [4], location-based services (LBS), cellular network-based indoor positioning [5], and so on.

Unmanned aircraft systems (UASs) are being used in several operational tasks and in civil, military, and scientific contexts, thanks to their flexibility, versatility, low cost, and ease of use, especially in environments that are very dangerous or impossible for human intervention [6]. UASs also show the potential to save lives, increase safety and efficiency, and enable more effective science and engineering research. Knowing the aircraft position in real time during a mission, particularly during beyond visual line of sight (BVLOS) or full autonomy operations, is essential [7,8,9]. As is well known, in outdoor operations, UAS positioning depends on onboard GNSS receivers that provide location coordinates in a geographical reference system. In many cases, the GNSS positioning accuracy (typically less than 5 m) is not sufficient for precision navigation (e.g., maneuvering in crowded areas, landing in impervious sites, or precision landing), due to attenuation, multipath errors, or signal reflections into urban canyons. On-board sensors and detect-and-avoid systems (DAA) can be used for supporting UAS navigation in harsh environments or sites where GNSS systems have low accuracy or in the absence of GNSS signals [10,11,12,13,14,15,16,17]. Moreover, positioning ability is limited in indoor scenarios or shelters because of blocked GNSS signals [18].

Indoor navigation systems have a wide range of applications: for example, wayfinding for humans in railway stations, bus stations, shopping malls, museums, airports, and libraries. Visually impaired individuals also could significantly benefit from indoor navigation systems. Needless to say, navigation through indoor spaces is more challenging [19]. Currently, UAS usage is not limited to outdoor operations, and unmanned aircraft can be useful in unknown or challenging indoor environments like hospitals, production companies, greenhouses, and manufacturing facilities. Many researchers are focusing their attention on the development of technical solutions for accurate and reliable UAS positioning during indoor operations, exploiting alternative technologies that must be compared to existing methods in terms of accuracy, precision, coverage, computational complexity, cost, and compatibility. Most of the solutions proposed in the literature are based on multi-sensor data fusion between INS/IMU (inertial navigation systems/inertial measurement unit) and other sensors such as camera, visual laser LiDAR, camera radar, LiDAR, or ultrasonic [20,21,22,23]. Alternative methodologies are based on image fusion with ultra-wide band (UWB) systems [24,25], on Wi-Fi access points (APs) [26], infrared (IR), radio frequency identification systems (RFID) [27], and computer vision [28]. Table 1 shows a comparison among typical technologies for indoor positioning [29] compared with our method.

In this work, we propose a low-cost UAS indoor positioning system based on Bluetooth low energy (BLE) beacons. A differential distance correction method (DDC) is proposed to improve the positioning accuracy. Sharing the same indoor propagation characteristics as 2.4-GHz Wi-Fi transceivers, BLE is the new specification of Bluetooth technology that guarantees transmission of small amounts of data and ultra-low power consumption [30,31]; consumption is up to 1% of the classic Bluetooth (with typical power consumption of 2.5 mW [32]. BLE beacons are a promising method for indoor positioning, especially in applications of position-based services, thanks to their low deployment cost, low power and dimensions, and suitability for a wide range of mobile devices. Table 2 shows some differences between classic Bluetooth and BLE.

The beaconing, or advertising, mode of BLE devices allows the hardware transmitter to broadcast advertising packets, i.e., short unsolicited messages, at flexible update rates [33]. Among the kinds of information permitted in the BLE standard, the received signal strength indicator (*RSSI*), which decreases with increasing distance, can be used to allow a receiving device to detect proximity to a specific BLE beacon based on the received signal strength (RSS). The flexibility of BLE device deployment can also allow good signal geometries for radio positioning; on the contrary, the location of Wi-Fi access points, typically near power sources, is chosen in order to maximize the signal coverage of the indoor area, rather than to optimize the wireless positioning of objects in the field.

BLE-based positioning methods typically consist of two approaches: range-based and fingerprint-based. The range-based method uses a predefined radio frequency (RF) path loss model to estimate the distance between the receiver (user) and the beacons. Assuming a minimum of three *RSSI* measurements, the user’s position can be solved by trilateration [34]. The fingerprint-based methods refer to the pattern of RSS measurements at a given location and consist of signal identity information (e.g., Wi-Fi MAC addresses or cellular IDs) and RSS values. Fingerprinting involves an offline and an online phase. During the offline phase, fingerprints at different places are collected to create a reference fingerprint map (RFM). In the online phase, a fingerprint collected at an unknown place is compared to the fingerprints in the RFM to solve for the user position [35].

This work applies to a range-based methodology by exploiting *RSSI* measurements for estimation of the distance between the receiver and a series of transmitters (beacons) using trilateration. To improve the accuracy, an extended Kalman filter (EKF) was employed on the noisy *RSSI* values. This study used a differential distance correction (DDC) methodology to correct the measured distances of a reference station from the beacons, and the position estimation of an object (rover) was calculated by trilateration and the *RSSI*-derived measurements. The differential correction was sent to the rover to attain a refined estimate of its location. During experimental tests, a Raspberry Pi 4 model B board was used as the reference (or master) station, and the Arduino Nano 33 BLE as the rover, whereas the anchor points (transmitters) were BLE Eddystone beacons. Experimental data were collected during several indoor tests conducted by the PFDL (Parthenope Flight Dynamics Labs) team of the University of Naples “Parthenope” (Italy).

The paper is organized as follows. In Section 2, the theoretical framework (*RSSI* distance method, trilateration method, EKF and DDC techniques) is presented. Section 3 describes the hardware used for the prototype of our positioning system. Simulations and results are shown and discussed in Section 4. Section 5 concludes the paper with final considerations and future work directions.

## 2. Theoretical Framework

The observation geometry of the BLE-based indoor positioning system (IPS) prototype mounted on a UAV is depicted in Figure 1. BLE was introduced in the Bluetooth 4.0 Standard, allowing advanced use for indoor localization technology by introducing a new type of device called “BLE beacons”. This new technology reduces costs and power consumption with respect to the “classical” Bluetooth; indeed, unlike the devices using the previous standard, the new ones have the option of transmitting at set intervals, which contributes significantly to the energy efficiency of the system, also improving the hardware and the immunity to interference. BLE has many similarities with Wi-Fi (in the 2.4-GHz band), and it is often used for indoor positioning in the same way as Wi-Fi, i.e., applying *RSSI*-based techniques. The BLE advertisement channels are labelled 37, 38, and 39 and are centered on 2402 MHz, 2426 MHz, and 2480 MHz, with 2 MHz bandwidth. Frequency hopping is used to communicate, each advertisement is repeated on each of the three channels, and the receiving device cycles over the advertising channels listening to the sent packets.

### 2.1. RSSI Distance Model

The principle of distance measurements by *RSSI* consists of transforming the signal attenuation into distance from the signal source, using the following empirical formula, which relates the BLE *RSSI* with the transmission distance based on the commonly used logarithmic attenuation model [36,37,38,39]:(1)$$L_{d}={\;L}_{l}+10\;n\log_{10}d+\nu$$
(2)$$L_{\;l}=10{\;log}_{10}\left[G_{t}G_{r}{\left(\frac{c/f}{4\pi }\right)}^{2}\right]$$
where $G_{t}$ and $G_{r}$ are the transmitting and receiver antenna gain, respectively; c is the speed of light; *f* is the carrier frequency; *n* is the channel attenuation coefficient (typically in a range from 2 to 6); ν is the noise (modeled as zero-mean, Gaussian, with variance ${\;\sigma }^{2}$, i.e., $N(0,{\;\sigma }^{2})$); *d* is the distance between receiver and transmitter; and $L_{d}$ is the channel loss after *d* meters. It is important to mention that the measurement error in *RSSI* does not regularly produce a Gaussian distribution, but for simplicity, the *RSSI* measurement error is treated as a Gaussian random variable [40,41]. A simplified relationship, based on Equation (1), in which we set the reference distance equal to 1 m, is [42]:(3)$$P_{r}\;(d)=A-10\;n\;{log}_{10}d$$
where $P_{r}$ represents the *RSSI* (in dBm); A (dBm) is the *RSSI* obtained when the signal transmits from 1 m distance from the receiver; and *n* is the propagation factor (also known as environmental factor or attenuation factor), which, as well as the noise standard deviation σ, depends on environmental conditions. When a BLE beacon is used, it periodically broadcasts an advertisement packet containing a unique ID and a calibrated *RSSI* value, relative to the distance with respect to receiver. This value allows us to determine the distance between a beacon and a device using the model in Equation (3), where *n* can also be interpreted as a calibration parameter for the path loss exponent. Obviously, the approach assumes that all installed beacons possess their predefined location information, including their exact coordinates. The general expression relating the *RSSI* to the distance is:(4)$$RSSI=\;A-10\;n\;log_{10}\left(\frac{d}{d_{0}}\right)$$
where $d_{0}$ is the reference distance value.

Time and space variations of the signal environment inevitably degrade the environmental factor of BLE reference nodes. When a BLE receiver (rover) enters the coverage area, there are not only reference nodes but also walls, other wireless devices, obstacles, and so on. The *n* factor describes the influence of walls and other obstacles inside the scenario. The environmental factor $n_{i}$ between two reference nodes can be estimated by:(5)$$n_{i}=\left(\frac{RSSI-A}{10\log_{10}d_{i}\;}\right)$$
where *RSSI* is the received power of the *i*-th reference node and $d_{i}$ is the known distance between the two reference nodes. At the reference distance $d_{0}$ (1 m), proximity and distance estimation can be calculated from Equation (4) as follows:(6)$$d=10^{\frac{A-RSSI}{10n}}$$

The localization algorithm can then be applied to use this distance and estimate the position using trilateration.

### 2.2. Extended Kalman Filter (EKF)

To obtain accurate position estimates from the ideal values of *RSSI*, filtering techniques are applied in order to mitigate noise effects and *RSSI* signal drift, resulting in better proximity estimation [43,44,45]. In this study, EKF was adopted for noise reduction and bias corrections considering the nonlinear characteristics of the beacon signal [46]. During the initial simulation tests, it was assumed that the UAV maintains a constant altitude (zero for simplicity). Consequently, dynamic behavior was equivalently characterized on a 2-D surface. EKF was designed to implement a measurement model, where the distance measurement based on the *RSSI* from the beacons exhibits a nonlinear relationship with the state values (UAS position). The measurement process model is
(7)$$x_{k+1}=f(x_{k},\;u_{k},\upsilon_{k})$$
(8)$$z_{k}=g(x_{k},\;u_{k},\;v_{k})$$
where k is the generic time index; x is the state; u is the input; *z* is the measured output; and υ and v are the process and measurement noises, respectively. We assume normal distribution for the noises $\upsilon_{k}$ and $v_{k}$, with known covariance matrices $Q_{k}$ and $R_{k}$:(9)$$\upsilon_{k}∼N\left(0,\;Q_{k}\right)\;\;\;\;\;\;v_{k}∼N(0,\;R_{k})$$

The algorithm is divided into two phases, prediction and correction. At each time step, the prediction phase provides the a priori state estimation ${\hat{x}}_{k+1}^{-}$ and the state covariance matrix $P_{k+1}^{-}$, whereas during the correction phase (update phase), the a posteriori state estimate ${\hat{x}}_{k+1}^{+}$ and the state covariance matrix $P_{k+1}^{+}$ are carried out.

The a priori estimation of the state ${\hat{x}}_{k+1}^{-}$ is performed by Equation (7), considering null process noise ${(\upsilon }_{k}=0):$(10)$${\hat{x}}_{k+1}^{-}=f\left({\hat{x}}_{k}^{+},\;u_{k},0\right)$$

The a priori state covariance matrix $P_{k+1}^{-}$ is obtained as follows:(11)$$P_{k+1}^{-}=A_{k}\;P_{k}^{+}\;A_{k}^{T}+W_{k}\;Q_{k}\;W_{k}^{T}$$
where
$$A_{k}={\left(\frac{\partial f}{\partial x}\right)}_{{\hat{x}}_{k}^{+},\;u_{k},0}\;\;\;W_{k}={\left(\frac{\partial f}{\partial \upsilon }\right)}_{{\hat{x}}_{k}^{+},\;u_{k},0}$$
are the Jacobian matrices of the function f, and the $P_{k}^{+}$ is the a posteriori state covariance matrix at time index *k*. Once a new measurement has been acquired, the a posteriori state estimation can be computed by updating the a priori state estimation:(12)$${\hat{x}}_{k}^{+}={\hat{x}}_{k}^{-}+K_{k}(z_{k}-h({\hat{x}}_{k}^{-}))$$
where the Kalman gain ($K_{k}$) is given by
(13)$$K_{k}=P_{k}^{-}\;H_{k}^{T}{\left(H_{k}P_{k}^{-}H_{k}^{T}+{V_{k}R}_{k}V_{k}^{T}\right)}^{-1}$$

$H_{k}$ is the observation matrix:$$H_{k}={\left(\frac{\partial z}{\partial x}\right)}_{{\hat{x}}_{k}^{-},\;u_{k},0}\;\;\;V_{k}={\left(\frac{\partial z}{\partial v}\right)}_{{\hat{x}}_{k}^{-},\;u_{k},0}$$

Finally, the a posteriori state matrix is updated as follows:(14)$$P_{k}^{+}=\left(I-K_{k}H_{k}\right)P_{k}^{-}$$
where I is the identity matrix.

### 2.3. Trilateration Method and Least-Squares Solution

Trilateration is a classical signal-based positioning technique that utilizes the estimated distances from known reference points to determine the target location [47]. At least three reference nodes are required to identify a unique solution. Each line of position is a circle with radius equal to the exact distance from the reference point, and the intersection of the circles is the target position, represented in Figure 2a for an ideal scenario. In real scenarios, errors in distance measurements will transform the intersection point into an overlap among the three circumferences, creating an uncertainty area (Figure 2b).

Consequently, there will be various possible solutions in the intersection area. When the number of distance measurements N is ≥3, the least squares method is adopted to calculate the optimal position of the target by minimizing the square of the offset in the user’s position relative to a linearization point (initial estimate of the user position), that is, the 2 × 1 vector $\delta \overset{⃑}{r}$, which is related to the *n* × 1 vector $\delta \overset{⃑}{d}$ (distance error, or offset in the distance values) by [48]
(15)$$\delta \overset{⃑}{d}=d_{i}-\hat{d_{ı}}=H\delta \overset{⃑}{r},\;\;i=1,\;\ldots ,N$$

The parameters present in Equation (15) are described below:(16)$$d_{i}=\sqrt{{\left(x-x_{i}\right)}^{2}+{\left(y-y_{i}\right)}^{2}}\;(\mathrm{real}\;\mathrm{beacon}\text{-}\mathrm{target}\;\mathrm{distance})$$
(17)$$\hat{d_{ı}}=\sqrt{{\left(\hat{x}-x_{i}\right)}^{2}+{\left(\hat{y}-y_{i}\right)}^{2}}\;(\mathrm{estimated}\;\mathrm{distance})$$
(18)$$H=\left[\begin{array}{cc} \frac{\hat{x}-x_{1}}{{\hat{d}}_{1}} & \frac{\hat{y}-y_{1}}{{\hat{d}}_{1}}\\ \frac{\hat{x}-x_{2}}{{\hat{d}}_{2}} & \frac{\hat{y}-y_{2}}{{\hat{d}}_{2}}\\ \vdots & \vdots \\ \frac{\hat{x}-x_{n}}{{\hat{d}}_{n}} & \frac{\hat{y}-y_{n}}{{\hat{d}}_{n}} \end{array}\right]$$
(19)$$\delta \;\overset{⃑}{r\;}=\left[\;\begin{array}{c} x-\;\hat{x}\\ y-\;\hat{y} \end{array}\right]=\left[\;\begin{array}{c} \delta x\\ \delta y \end{array}\;\right]$$
where ${\left(x,y\right)}^{T}$ are the real target coordinates, or user’s position, where the superscript *T* stands for transpose; ${\left(x_{i},y_{i}\right)}^{T}$ are the known *i*-th beacon coordinates; ${\left(\hat{x},\hat{y}\right)}^{T}$ are the estimated target coordinates; and H is the $\in R^{Nx2}$ design matrix derived by linearizing Equation (18) about an initial position guess. The least-squares solution provides the offset of the user’s position from the linearization point expressed as a linear function of $\delta \overset{⃑}{d}$:(20)$$\delta \overset{⃑}{r}={\left(H^{T}\;H\right)}^{-1}\;H^{T}\;\delta \overset{⃑}{d}$$

Once $\delta \overset{⃑}{r}$ is calculated, a new estimate of the user’s position is obtained from Equation (19), and a linearization around the new estimate is performed, obtaining the optimal solution by the iteration
(21)$${\left[\begin{array}{c} \;\hat{x}\\ \;\hat{y} \end{array}\;\right]}_{k+1}={\left[\begin{array}{c} \;\hat{x}\;\\ \hat{y} \end{array}\right]}_{k}+{\left[\;\begin{array}{c} \delta x\\ \delta y \end{array}\;\right]}_{k}$$
where *k +* 1 denotes the updated coordinate vector and *k* denotes the vector at the previous iteration. The acceptable displacement ${\left(\delta x,\delta y\right)}^{T}$ is related to the accuracy requirements. The 3-D positioning by trilateration allows one to find the user position using the *RSSI* signal received from at least four non-coplanar beacons at known positions (the fourth equation removes the ambiguity between two possible solutions given by three spheres intersecting in two points):(22)$$\left\{\begin{array}{c} {\left(x-x_{1}\right)}^{2}+{\left(y-y_{1}\right)}^{2}+{\left(z-z_{1}\right)}^{2}={\;d}_{1}^{2}\\ {\left(x-x_{2}\right)}^{2}+{\left(y-y_{2}\right)}^{2}+{\;\left(z-z_{2}\right)}^{2}={\;d}_{2}^{2}\\ {\left(x-x_{3}\right)}^{2}+{\;\left(y-y_{3}\right)}^{2}+{\left(z-z_{3}\right)}^{2}=d_{3}^{2}\\ {\left(x-x_{4}\right)}^{2}+{\;\left(y-y_{4}\right)}^{2}+{\left(z-z_{4}\right)}^{2}=d_{4}^{2} \end{array}\right.$$
where (*x*, *y*, *z*) are the unknown 3-D coordinates of the target Bluetooth receiver.

During preliminary simulation tests, the *z* coordinates were set to zero to evaluate the accuracy and stability of the method. Considering 2-D positioning, it is possible to obtain the (*x*, *y*) coordinates of the receiver by
(23)$$x=\frac{\left(x_{1}^{2}+y_{1}^{2}-d_{1}^{2}\right)\left(y_{3}-y_{2}\right)+\left(x_{2}^{2}+y_{2}^{2}-d_{2}^{2}\right)\left(y_{1}-y_{3}\right)+(x_{3}^{2}+y_{3}^{2}-d_{3}^{2})(y_{2}-y_{1})}{2[y_{1}\left(x_{3}-x_{2}\right)+y_{2}\left(x_{1}-x_{3}\right)+y_{3}(x_{2}-x_{1})]}$$
(24)$$y=\frac{\left(x_{1}^{2}+y_{1}^{2}-d_{1}^{2}\right)\left(x_{3}-x_{2}\right)+\left(x_{2}^{2}+y_{2}^{2}-d_{2}^{2}\right)\left(x_{1}-x_{3}\right)+(x_{3}^{2}+y_{3}^{2}-d_{3}^{2})(x_{2}-x_{1})}{2[y_{1}\left(x_{3}-x_{2}\right)+y_{2}\left(x_{1}-x_{3}\right)+y_{3}(x_{2}-x_{1})]}$$

With the known locations of the beacons $B1=\left(0,0\right);B2=\left(0,y_{2}\right);B3=(x_{3},0)$, as shown in Figure 3, Equations (23) and (24) are simplified as follows [49]:(25)$$x=\frac{x_{3}^{2}+(d_{1}^{2}-{\;d}_{2}^{2})}{2x_{3}}$$
(26)$$y=\frac{y_{2}^{2}+(d_{1\;}^{2}-d_{3}^{2})}{2y_{2}}$$

### 2.4. Differential Distance Correction Technique

The distance converted from *RSSI* is an estimated value influenced by environmental issues such as multipath, reflections, and shadowing. The DDC methodology uses multiple reference stations to estimate the distance error [50]. The concept is similar to differential GNSS (DGNSS) and real-time kinematic (RTK) positioning that use the differential information from the reference station (Master), i.e., the error $\delta \overset{⃑}{r\;}$, to correct the position of the rover, in the hypothesis that the measurements performed by the master station(s) and the rover are affected by the same environmental errors (i.e., with low temporal and spatial latency). Figure 4 shows the schematic diagram of differential correction [51,52] with four beacons located at known positions. The coordinates of the master station are also known in advance.

The master station compares the estimated position to its known coordinates to evaluate the errors and sends the corrections to be applied to the *RSSI* values measured by the rover. To estimate the residual at every location in the field, the inverse distance weighted (IDW) interpolation method is adopted. The estimated residual $r_{est}$ at unknown coordinate (*x*, *y*) by IDW can be calculated from the equation as follows [50,52]:(27)$$r_{est}=\sum_{i=1}^{N}w_{i}\times r_{i}$$
where $r_{est}$ is the estimated residual from IDW, $r_{i}$ is the estimated residual of the *i*-th reference station, and $w_{i}$ is the weight of the *i*-th reference station.

## 3. Hardware Description

The prototype of our experimental BLE-based IPS is composed of

Raspberry Pi 4 model B is used as the master station.Arduino Nano 33 BLE used as the rover.BLE Eddystone beacons as transmitting nodes.

### 3.1. Master Station: Raspberry Pi 4 Model B

The Raspberry Pi 4 Model B used for our prototype system (Figure 5) is a board with features like a camera connector, 802.11ac Wi-Fi, Bluetooth 5, full gigabit Ethernet (throughput not limited), two USB 2.0 ports, two USB 3.0 ports, 1–8 GB of RAM, dual-monitor support via a pair of micro-HDMI (HDMI type D) portsFirst bullet, GPIO pins for interfacing sensors and switches, USB ports to connect to external devices (keyboard, mouse, Wi-Fi adapter, etc.), and an audio jack [53]. The board has no internal mass storage or built-in operating system, requiring an SD card preloaded with a version of the Linux Operating System. Due to its relatively low current consumption, Raspberry Pi can be powered using a 5 V USB Power Bank, and it can be carried around by a subject or placed on the object to be tracked. The main algorithm is hosted on the 1.5 GHz 64-bit quad core ARM Cortex-A72 processor on-board the Pi 4.

The master station is connected to a PC-based ground control station (GCS) by hotspot connection to send and store *RSSI* values of each single transmitter. The main task of the Raspberry Pi microcomputer is to receive and process the messages transmitted by the BLE beacons, in particular, the beacon identifier address and the *RSSI* value. From these values, the algorithm estimates the real-time position and compares it to its real position in order to calculate the error (residual) and send the correction to the rover.

### 3.2. Rover: Arduino Nano 33 BLE

Arduino Nano 33 BLE is a multiprotocol microcontroller (Figure 6, Table 3), supporting only 3.3 V I/Os. In addition, the 5 V pin is connected, through a jumper, to the USB power input. There are two 15-pin connectors on the board, one on each side, pin to pin compatible with the original Arduino Nano. The main processor is the nRF52840 (Nordic Semiconductors), a 32-bit ARM Cortex-M4 running at up to 64 MHz. The main processor includes other features like Bluetooth pairing via NFC (near field communication) and ultra-low-power-consumption modes. It has an embedded 9-axis inertial sensor that makes this board ideal for wearable devices but also for a large range of scientific experiments in the need of short-distance wireless communication. Most of its pins are connected to the external headers; however, some are reserved for internal communication with the wireless module and the on-board internal I^2^C peripherals (IMU and Crypto) [54,55].

The Arduino nano 33 BLE is powered by two 2600-mAh Li-ion 18650 3.7-V batteries and has a microSD connected to the board to store *RSSI* values. Figure 7 shows the prototype of the rover station. The algorithm running on the Arduino can receive data from the beacons and save them on the micro-SD.

### 3.3. Transmitters: BLE Eddystone Beacons

A beacon is a hardware transmitter capable of broadcasting periodically a short message to nearby portable electronic devices. The basic information of the advertising packet from the beacon is shown in Table 4.

We used Eddystone BLE beacons with an embedded nRF51822 chip (Figure 8). Eddystone is a protocol specification that defines a BLE message format for proximity beacon messages. It describes several different frame types that may be used individually or in combinations to create beacons that can be used for a variety of applications [56].

The nRF51822 is a powerful multi-protocol single chip solution for wireless applications. It incorporates a 32-bit ARM Cortex M0 CPU and 256 kB flash + 16 kB RAM memory, and it supports Bluetooth low energy and 2.4 GHz protocol stacks [57]. The programmable peripheral interconnect (PPI) system provides a 16-channel bus for direct and autonomous system peripheral communication without CPU intervention. This brings predictable latency times for peripheral-to-peripheral interaction and power saving benefits associated with leaving the CPU idle.

## 4. Simulation and Results

The software for IPS is divided into three phases, shown in Figure 9, each running on its own platform. MATLAB 2024b^®^ runs on the GCS and is used for non-real-time algorithms (post-processing).

### 4.1. Calibration Phase

During the first phase, environment calibration is performed, necessary for environmental factor (*n*) modelling and Kalman filter calibration. Preliminary data collection consisted of indoor *RSSI* measurements at known distances between the beacons and the receiver. The distance range considered was 0.25–5 m, with steps of 0.25 m for the range 0.25–3 m, and with steps of 0.50 m for the range 3–5 m. The acquisition time was about 10 min in static conditions. The measurement procedure was performed in a clean environment (without metallic objects and the presence of other nearby devices). This first phase (environmental calibration, *n* factor modeling and Kalman filter calibration) is presented in [58], where the data collections and preliminary results are shown in detail. Noting the high presence of instability in the *RSSI* signal, a simple smoothing filter based on a moving average was applied before implementing the EKF algorithm. Figure 10 and Figure 11 show data collections (raw *RSSI* values and smoothing filtered data), during the calibration phase, where the distances between receiver and transmitters (four beacons were used) were 3 and 0.75 m, respectively. Figure 12 shows the comparison between variances of the raw and filtered *RSSI* values, and Figure 13 shows the *RSSI* mean values of the filtered data.

After data collection, the environmental factor *n* was evaluated for each known distance and for each beacon, following Equation (5). This factor describes the influence of walls and other obstacles inside the scenario (Figure 14).

### 4.2. Experimental Tests: 2-D Scenario

After the calibration phase, experimental tests were conducted by positioning the beacons and the master station within a designated area. The rover, simulating a UAV maintaining a constant altitude (in this case equal to zero simulating take-off positioning area, reducing dynamic behavior on a 2-D surface), was placed at unknown distances from the transmitters and the master station within this area. Experimental indoor tests were performed in a free area inside the University of Studies of Naples “Parthenope” (Italy). The field has a length of 3 m and width of 3 m. The arrangement of the experimental field is depicted in Figure 15, where there were four beacons (B1, …, B4) and one reference station in the field. Table 5 shows the coordinates of the work area, delimited by the beacon positions, and of the master station.

During the static tests, the master station was connected to the GCS by hotspot connection to store *RSSI* data from the beacons, whereas the rover was equipped with a micro-SD, allowing *RSSI* data storage. The acquisition time was about 10 min.

In post-processing, the raw master receivers’ coordinates were calculated following Equations (25) and (26), where $d_{1},\;d_{2}$ and $d_{3}$ were the unknown distances from B1, B2, and B3, respectively (B4 was added in case of failure of one of the other beacons). These distances were derived from Equation (6), where *n* (see Figure 14) was obtained by using Equation (5). Figure 16 shows the raw coordinates of the master station compared to its actual position.

In the successive step, EKF was applied to mitigate noise, bias, and drift issues. Table 6 shows the parameters used during the EKF initialization phase.

Figure 17 and Figure 18 show a comparison between raw data, EKF data, and the actual position of the master station and rover station, respectively.

Table 7 presents the real positions of the master and rover stations, along with their corresponding raw and EKF coordinate mean values.

Following these steps, the coordinates of the master station were calculated and compared to the real coordinates to evaluate the residual (correction), used to correct the rover position (Figure 19). Then, the rover coordinates were calculated, and the residual was added, correcting its position.

Figure 20 shows the configuration area and the comparison between the testing point’s estimated position and the real position of the rover.

Table 8 compares raw and filtered data in terms of the standard deviation of the error.

Table 9 shows the results indicating the final position corrected using the DDC method, in terms of mean value.

The second test was conducted using the same configuration area, beacons, and master positions as the first test, but with a different rover position. Figure 21 illustrates the configuration area, and Table 10 presents the obtained results.

### 4.3. Experimental Test: 3-D Scenario

In this section, a 3-D test is presented, considering the same scenario depicted during the tests performed and described in the previous section. For this test, the rover station was positioned on a vertical structure, at 1.50 m above the ground (as shown in Figure 22), simulating a drone in hovering phase.

The beacons were positioned at the same coordinates as in the 2-D tests. To include the third dimension, they were placed one meter above the ground. The master station’s coordinates remained similar, simulating a ground station. Table 11 shows the coordinates of the work area during the 3-D test, whereas Table 12 shows the parameters used during the EKF initialization phase for the 3-D scenario.

Figure 23 depicts the filtered coordinates of the master station, which will be compared to its real position to calculate the residual and subsequently correct the rover’s position, as illustrated in Figure 24.

Table 13 shows the results indicating the final position corrected using the DDC method in terms of mean value.

## 5. Conclusions and Further Work

This research proposes a preliminary indoor positioning system prototype utilizing BLE technology for indoor missions using small UAVs. The methodology is based on the *RSSI* measured from BLE beacons and on trilateration. To improve the performance of the positioning method, extended Kalman filtering (EKF) and the differential distance correction (DDC) method were used. The DDC technique, similar to differential GNSS and RTK positioning, uses differential information from the reference station to correct the position of the rover station. The reference station (master) in a known position computes the residual of distance, sending a correction to be applied to the distance observation made by the receiving station (rover). The DDC-corrected distance value is used in the trilateration scheme to obtain a better estimate of the rover’s position. In order to reduce noise effects, bias, and *RSSI* signal drift, EKF was applied on raw data. The components of the IPS prototype consist of the transmitters (Eddystone BLE beacons) placed in the indoor operating field, a master station (Raspberry Pi 4 model B), and an autonomous rover station (Arduino Nano 33 BLE), which will be mounted on-board the UAV, added to the standard instrumentation, to enhance the UAS position during indoor operations.

A first analysis was performed to verify the device characteristics and the functionality and accuracy of the methodologies applied, considering only 2-D positioning in static conditions. The most critical aspect was found to be the accurate modeling of the environmental factor, which can cause signal fluctuations due to multipath, increasing the measurement errors. To reduce multipath effects, reflection, and fading, the tests were carried out in a clean environment (without metallic objects and other devices in the test area). The results demonstrate an improved position estimate after the DDC correction. As shown in Table 9, the estimated residual errors were 0.48 m on the x-coordinate and 0.01 m on the y-coordinate. The rover position estimate improved from an error of 27.5% and 29.5% for the x- and y-coordinates, respectively, before DDC correction (Table 7), to an error of 24% on the x-coordinate and 0.5% on the y-coordinate after the application of the EKF and DDC methodology during test 1. Additionally, an error of 2.7% on the x-coordinate and 14.7% on the y-coordinate was obtained during the second test. Finally, a test considering a 3-D scenario was performed. In this test, a hovering phase within a terminal area was simulated by positioning the rover prototype on a vertical structure at 1.50 m above the ground. Following the same process described during the 2-D tests, the application of EKF and DDC methods resulted in an error of 25% on the x-coordinate, 16.5% on the y-coordinate, and 18.6% on the z-coordinate.

Further research will focus on conducting other experiments in an empty room, as well as new tests with obstacles and a greater number of beacons and master stations, performing 3-D positioning in dynamic conditions, to compare the influence of the obstacles in the experimental environment, and mounting the rover device on-board a UAV. Based on these promising results, our future objective is to expand coverage to larger areas by implementing a “mosaic” strategy. This would entail covering extensive areas by creating multiple smaller, controlled zones similar to the test area used in this study, ensuring consistent performance across a broader field. Additionally, future developments will concern analysis of the accuracy of BLE *RSSI* values as a function of the environmental changes (for example, considering more targets, changes in the density of people present in the test area, position of furniture or walls, and changes that will require regular re-calibration of the IPS to ensure the accuracy of the methodology), as well as the study of BLE data fusion with other systems, for example, Wi-Fi for increasing the number of reference nodes, or on-board devices as IMUs to reduce positioning errors. In the future, the proposed IPS will be integrated into a safe landing area determination system recently developed by the authors [17].

## Figures and Tables

**Figure 1 sensors-24-07170-f001:**
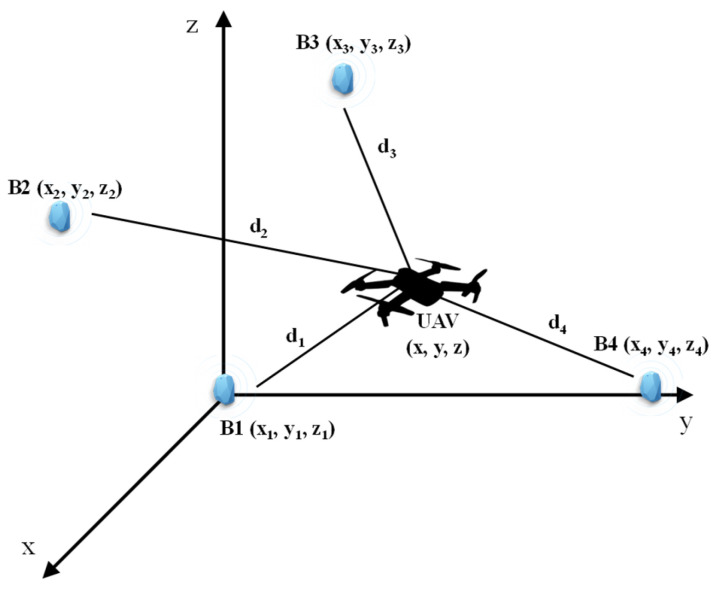
UAS indoor 3-D positioning system with four BLE devices. The ideal aircraft position is provided by the intersection of four spheres with centers on the known positions of the beacons B1, …, B4.

**Figure 2 sensors-24-07170-f002:**
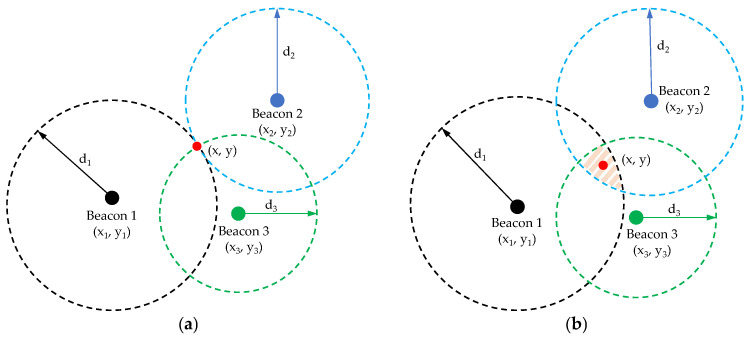
The 2-D trilateration method in an ideal (**a**) and a real (**b**) scenario.

**Figure 3 sensors-24-07170-f003:**
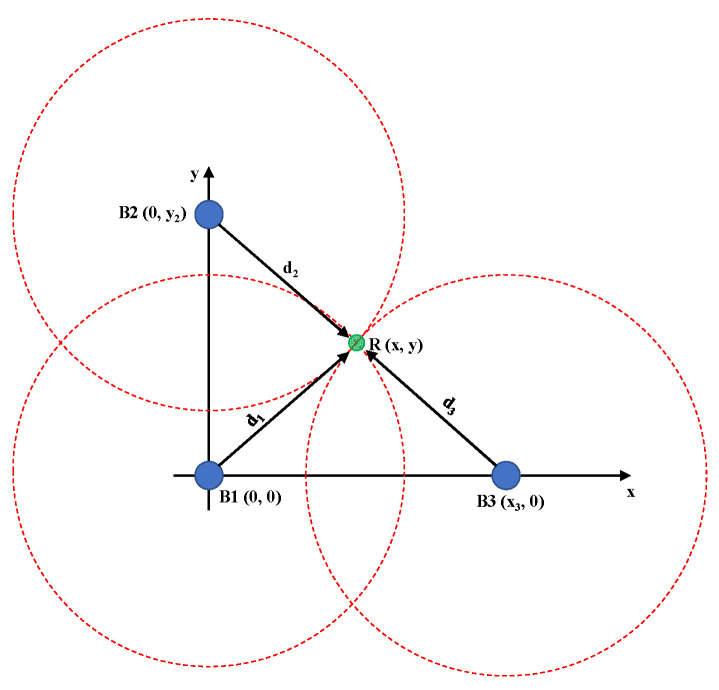
Positioning based on 2-D trilateration with three BLE beacons.

**Figure 4 sensors-24-07170-f004:**
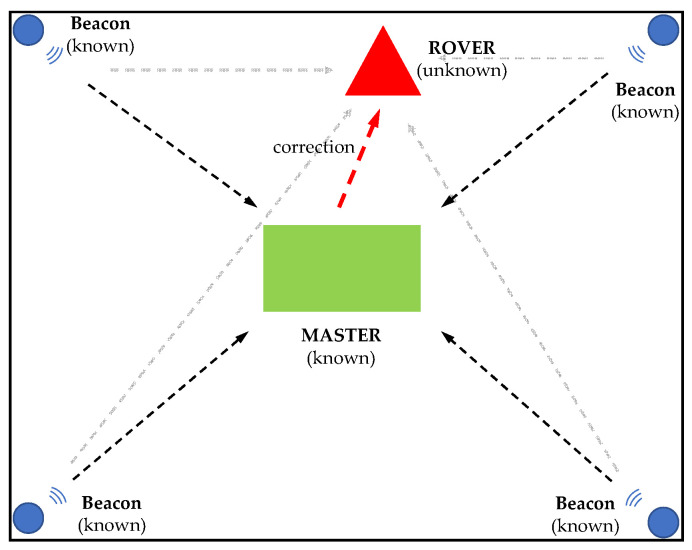
Schematic diagram of DDC methodology with a single master station of known position.

**Figure 5 sensors-24-07170-f005:**
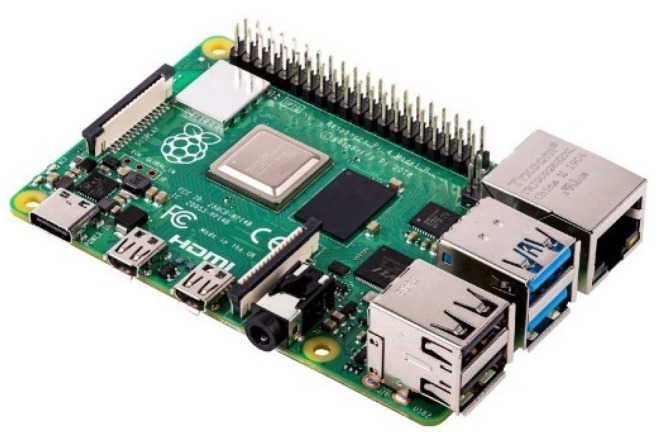
Raspberry Pi 4 model B.

**Figure 6 sensors-24-07170-f006:**
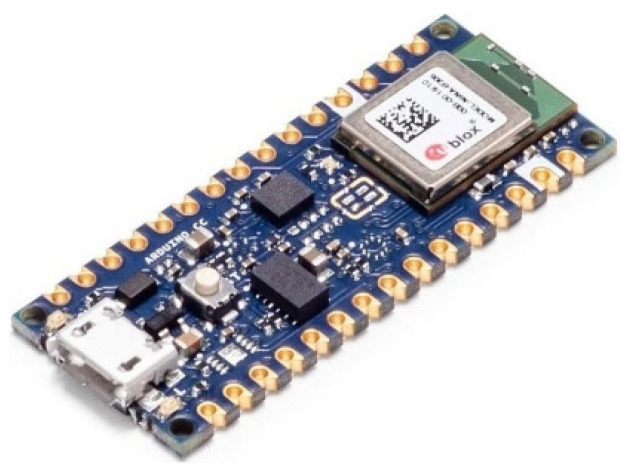
Arduino nano 33 BLE.

**Figure 7 sensors-24-07170-f007:**
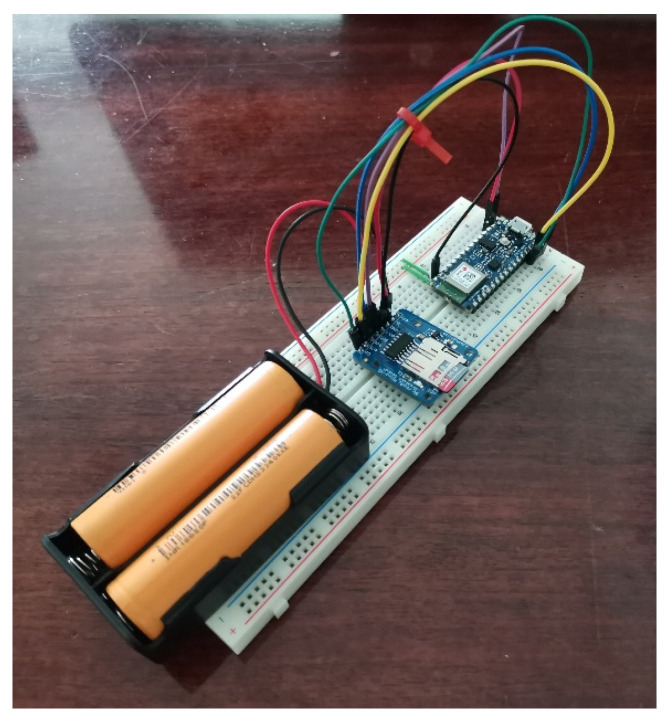
Rover station prototype.

**Figure 8 sensors-24-07170-f008:**
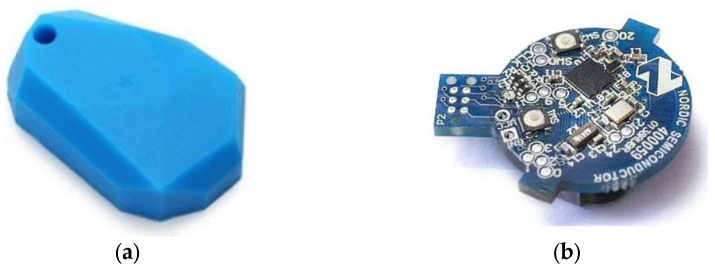
Eddystone beacon: (**a**) silicon cover, (**b**) chip nRF51822.

**Figure 9 sensors-24-07170-f009:**
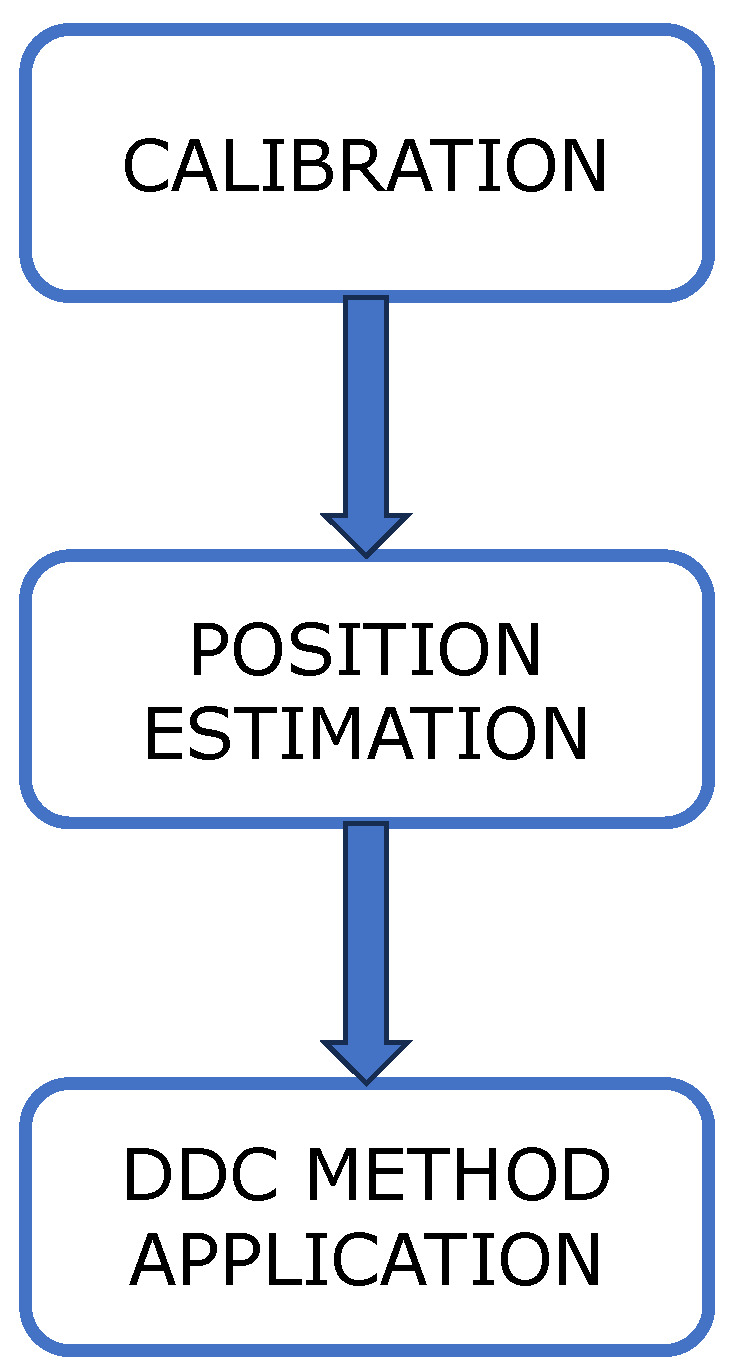
Phases of the proposed IPS.

**Figure 10 sensors-24-07170-f010:**
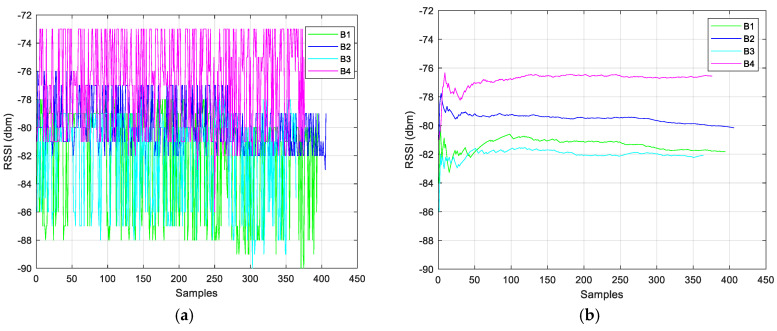
*RSSI* values, raw (**a**) and filtered (**b**), at 3 m distance.

**Figure 11 sensors-24-07170-f011:**
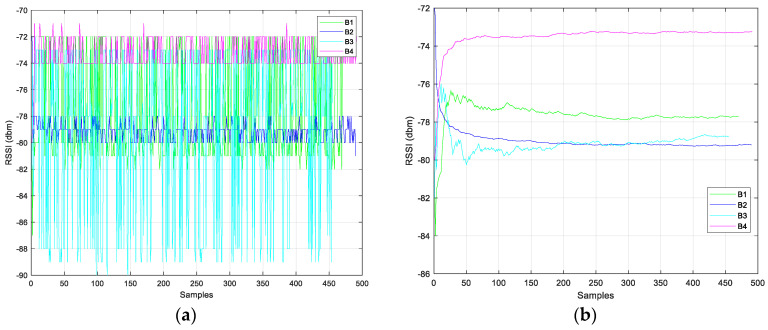
*RSSI* values, raw (**a**) and filtered (**b**), at 0.75 m.

**Figure 12 sensors-24-07170-f012:**
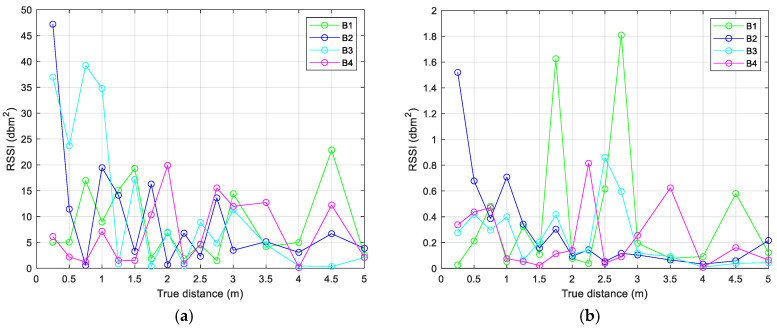
Variance of the *RSSI* raw (**a**) and filtered (**b**) values.

**Figure 13 sensors-24-07170-f013:**
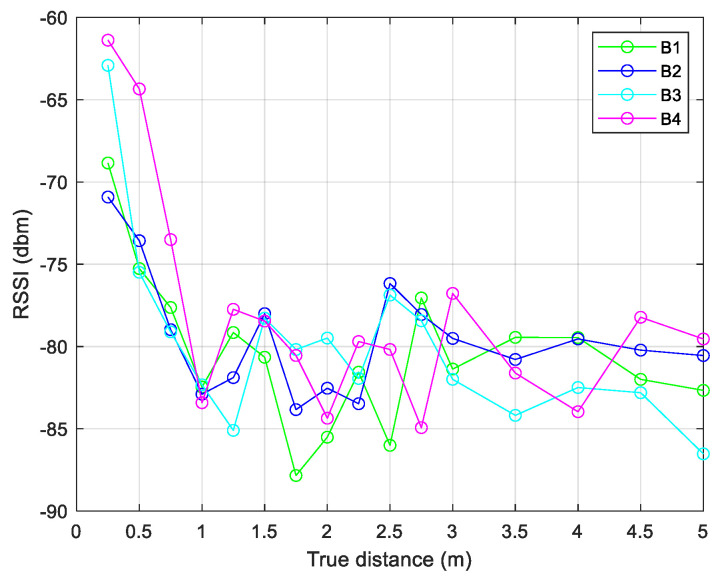
Mean *RSSI* values of the filtered data.

**Figure 14 sensors-24-07170-f014:**
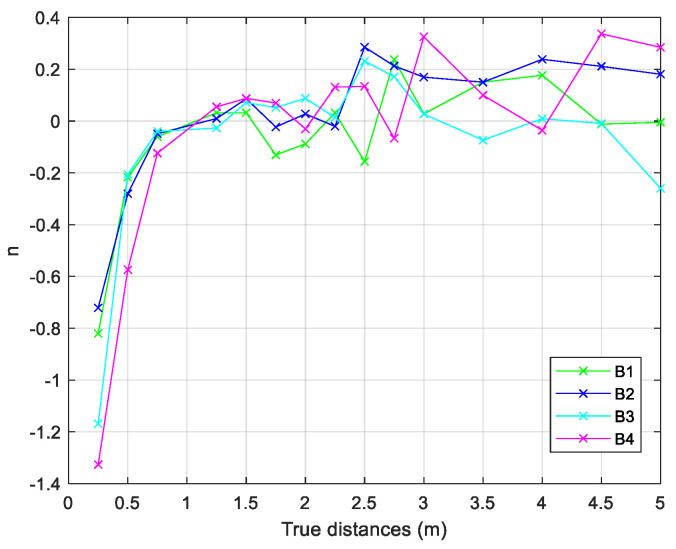
Trend of the measured environmental factor.

**Figure 15 sensors-24-07170-f015:**
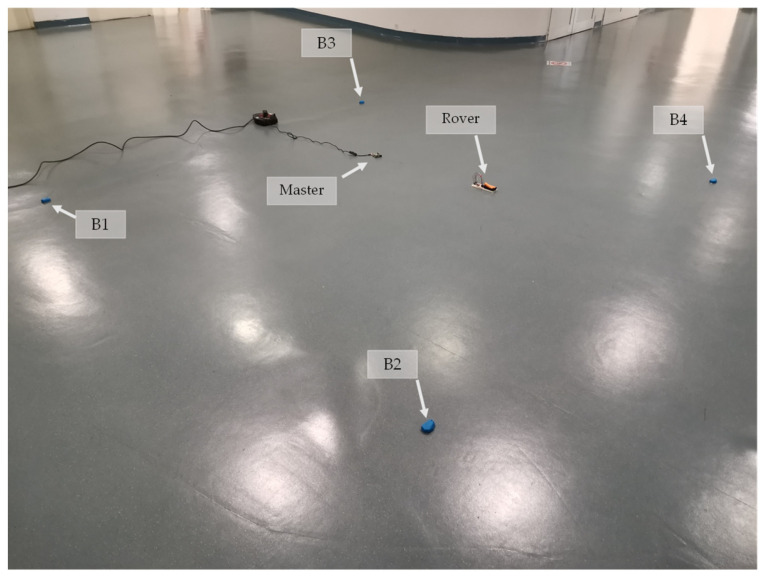
Experimental setup area.

**Figure 16 sensors-24-07170-f016:**
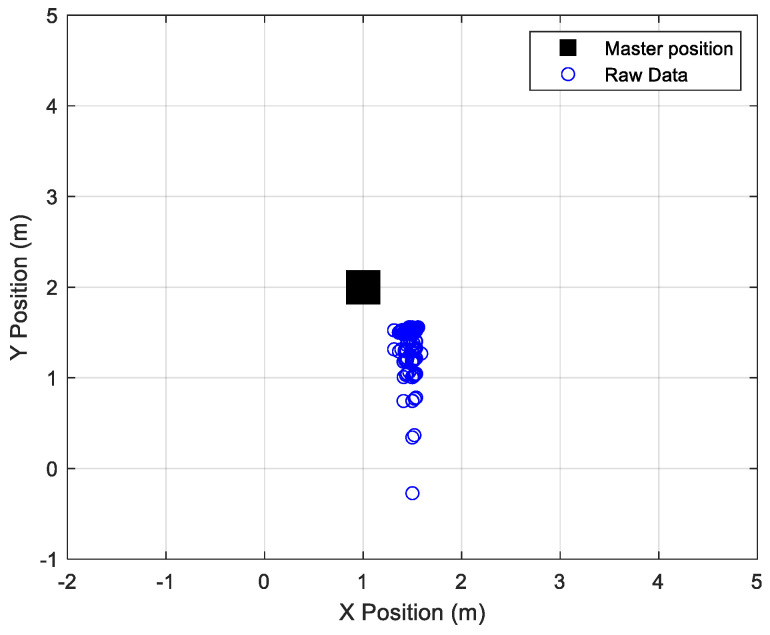
Comparison among real and raw master coordinates.

**Figure 17 sensors-24-07170-f017:**
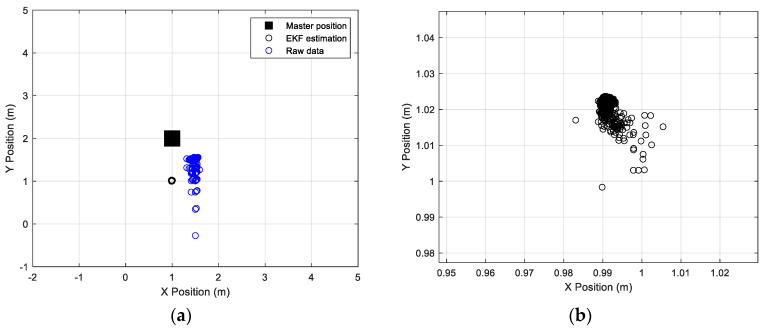
(**a**) Raw and EKF coordinates estimation, compared to the real position of the master station. (**b**) Zoom view of the EKF data estimation.

**Figure 18 sensors-24-07170-f018:**
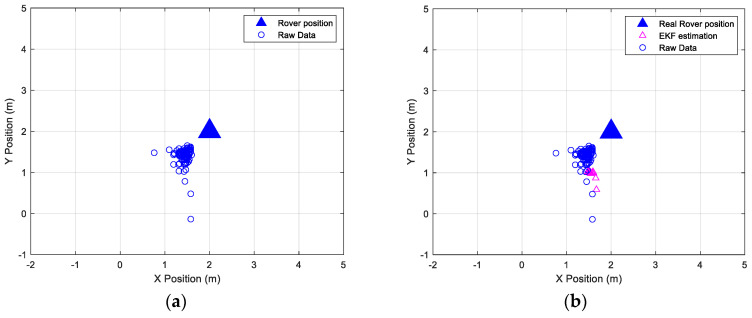
(**a**) Raw and EKF coordinates estimation, compared with the real position of the rover station. (**b**) Zoom view of the EKF data estimation.

**Figure 19 sensors-24-07170-f019:**
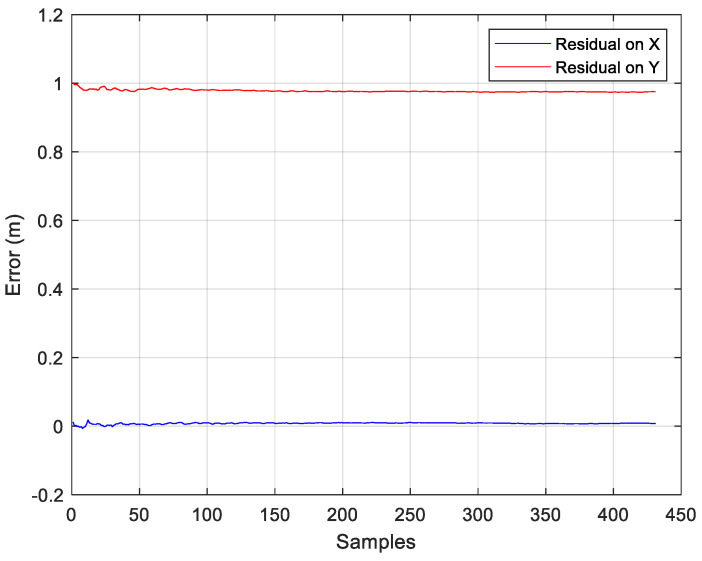
Error calculated on known master coordinates, used to correct the rover position by the DDC method.

**Figure 20 sensors-24-07170-f020:**
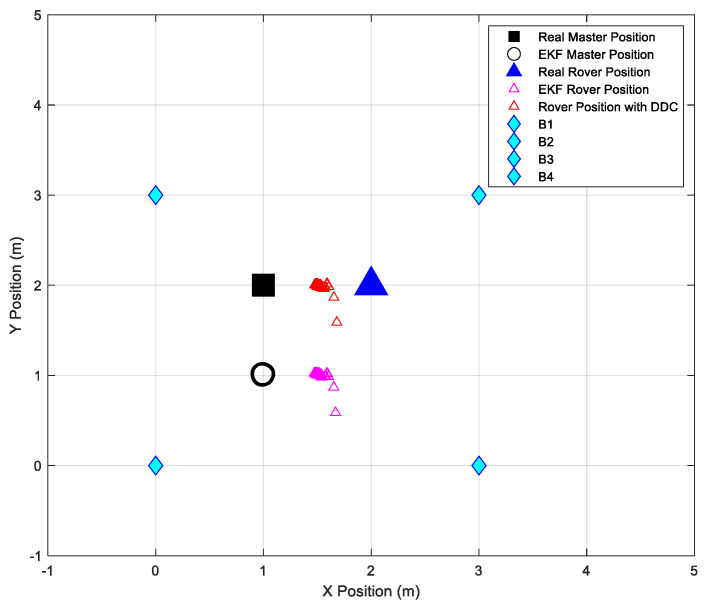
Configuration area representing final positioning results.

**Figure 21 sensors-24-07170-f021:**
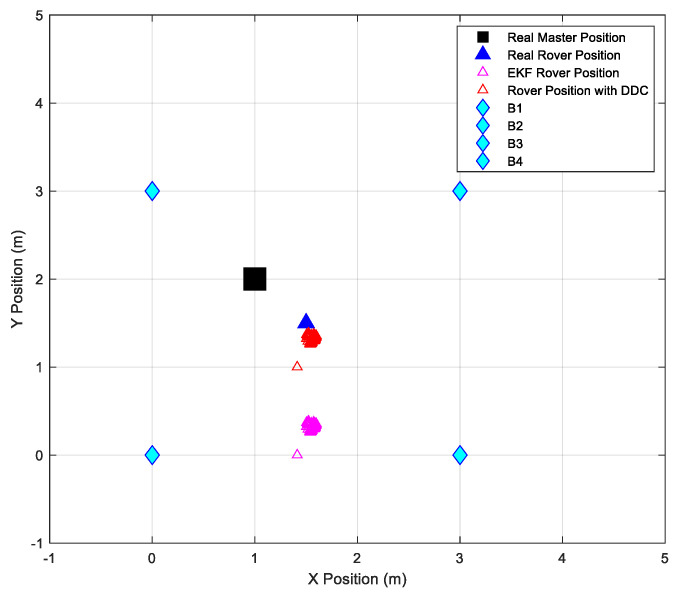
Configuration area representing final positioning results during the second test.

**Figure 22 sensors-24-07170-f022:**
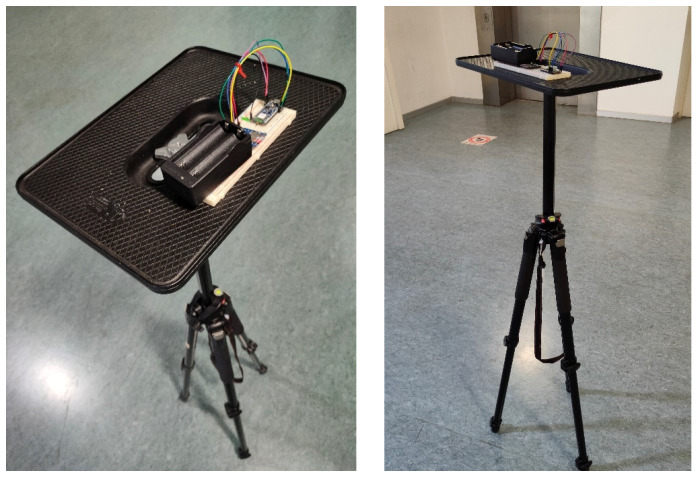
Rover prototype placed on a vertical structure.

**Figure 23 sensors-24-07170-f023:**
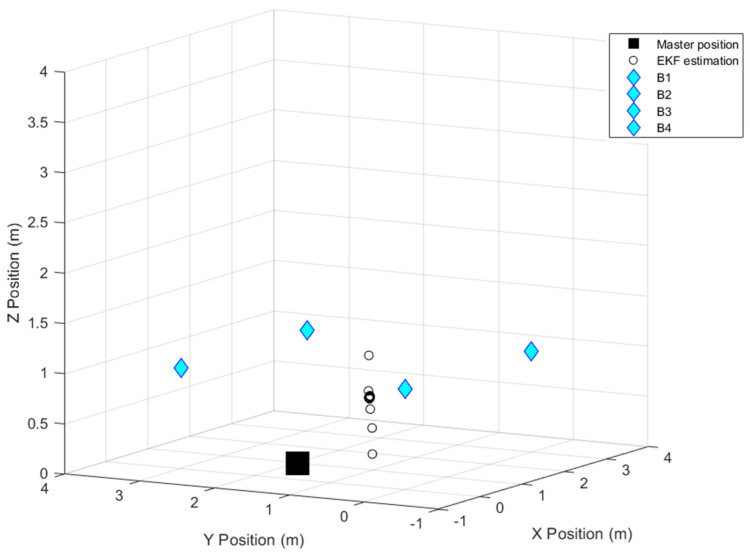
EKF data estimation of the master station in the 3-D scenario.

**Figure 24 sensors-24-07170-f024:**
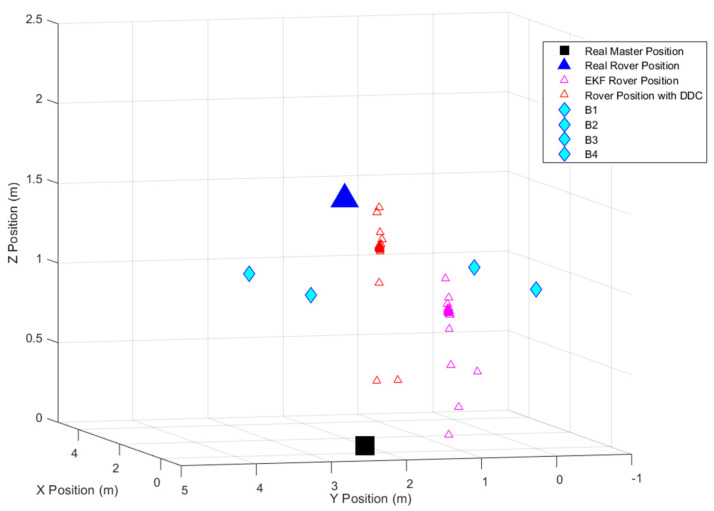
Configuration area representing final positioning results during the 3-D scenario.

**Table 1 sensors-24-07170-t001:** Overview of indoor positioning technologies.

Technology	Accuracy (m)	Power Consumption	Extra Device	Cost
Wi-Fi	~2–5	High	No	Low
Bluetooth	2–5	Low	No	Low
BLE	1–5	Very low	No	Very low
RFID	~1–3	Low	Yes	Moderate
UWB	~0.1–0.5	Low	Yes	High
Infrared	0.5–3	Low	Yes	Moderate
Acoustic signal	0.3–0.8	Low	No	Moderate
BLE + DDC (our method)	0.4–0.6	Very low	Yes	Very low

**Table 2 sensors-24-07170-t002:** Classic Bluetooth vs. BLE technology.

Propriety	Bluetooth	BLE
Frequency	2.4 GHz	2.4 GHz
Data rate	1 to 3 Mbps	1 Mbps
Range	Up to 10 m	Up to 40 m
Power consumption	Low	Very low
Battery life	Multiple weeks	Multiple months
Cost	Low	Very low
Accuracy	2–5 m	1–5 m

**Table 3 sensors-24-07170-t003:** Arduino nano 33 BLE main characteristics.

Type Characteristics	Nano 33 BLE
Microcontroller	nRF52480
Clock speed	64 MHz
Flash	1 MB
RAM	256 KB
Connectivity	BLE
Sensors	9-axis IMU

**Table 4 sensors-24-07170-t004:** Beacon advertising packet [51].

Beacon prefix	Fixed by beacon protocol and makes information follow the protocol.
UUID identifier	Identifier that should be used to distinguish the different classes of beacons in a wide range.
Major	Identifier for determining a different group of beacons.
Minor	Identifier for determining individual beacons.
*RSSI*	The radio signal strength indicator (*RSSI*) is a measurement of the power present in a received radio signal.
Measured power	The *RSSI* value, which is measured at 1 m away from a beacon.

**Table 5 sensors-24-07170-t005:** Work area: beacon and maser coordinates (origin in B1 position).

Devices	Coordinates (m)
Master	(1, 2)
B1	(0, 0)
B2	(3, 0)
B3	(0, 3)
B4	(3, 3)

**Table 6 sensors-24-07170-t006:** EKF initial parameters.

Parameter	Value
Initialization state x	[0, 0]
$\sigma_{R}^{2}$ (dbm^2^)	2
$\sigma_{Q}^{2}$ (dbm^2^)	1
P	I(2)
R	$\sigma_{R}^{2}\;$∙ I(4)
Q	$\sigma_{Q}^{2}\;$∙ I(2)

**Table 7 sensors-24-07170-t007:** Comparison among real, raw, and EKF coordinates of the master station and rover.

Master Station Coordinates			Rover Coordinates		
Real (m)	Raw (m)	EKF (m)	Real (m)	Raw (m)	EKF (m)
(1, 2)	(1.48, 1.37)	(0.99, 1.02)	(2, 2)	(1.45, 1.41)	(1.51, 1.01)

**Table 8 sensors-24-07170-t008:** Error standard deviation (raw and EKF data).

Master Station		Rover	
STD Raw Data (m)	STD EKF (m)	STD Raw Data (m)	STD EKF (m)
(0.05, 0.59)	(0.002, 0.003)	(0.10, 0.18)	(0.02, 0.03)

**Table 9 sensors-24-07170-t009:** Positioning results with DDC methodology during the first test.

Rover Coordinates	
Real (m)	With DDC (m)
(2, 2)	(1.52, 1.99)

**Table 10 sensors-24-07170-t010:** Positioning results with the DDC method during the second test.

Rover Coordinates	
Real (m)	With DDC (m)
(1.5, 1.5)	(1.54, 1.28)

**Table 11 sensors-24-07170-t011:** Work area: beacon and maser coordinates (origin in B1 position) during the 3-D test.

Devices	Coordinates (m)
Master	(1, 2, 0)
B1	(0, 0, 1)
B2	(3, 0, 1)
B3	(0, 3, 1)
B4	(3, 3, 1)

**Table 12 sensors-24-07170-t012:** EKF initial parameters.

Parameter	Value
Initialing state x	[0, 0, 0]
$\sigma_{R}^{2}$ (dbm^2^)	2
$\sigma_{Q}^{2}$ (dbm^2^)	1
P	I(3)
R	$\sigma_{R}^{2}$∙ I(4)
Q	$\sigma_{Q}^{2}$∙ I(3)

**Table 13 sensors-24-07170-t013:** Comparison among real, EKF, and DDC methods for the rover coordinates.

Rover Coordinates		
Real (m)	EKF (m)	With DDC (m)
(2, 2, 1.5)	(1.54, 0.74, 0.81)	(1.50, 1.67, 1.22)

## Data Availability

Data are contained within the article.

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
