# Peer review of "Differential Positioning with Bluetooth Low Energy (BLE) Beacons for UAS Indoor Operations: Analysis and Results"

_sensors, 2024, doi:10.3390/s24227170_

Round 1

Reviewer 1 Report (Previous Reviewer 1)

Comments and Suggestions for Authors

The authors have addressed my concerns. I have no further questions.

Author Response

Reviewer #1:

The authors have addressed my concerns. I have no further questions.

The authors thank the Reviewer for their careful review of our paper.

Reviewer 2 Report (New Reviewer)

Comments and Suggestions for Authors

The methodology of determining UASs' position using low-energy Bluetooth beacons is presented in the paper. The article is well-structured. The experimental data for 2D and 3D position systems are presented (but in static regime only). The references cited in this paper appropriate and relevant to this research.  Some comments:

1) It is not entirely clear why the accuracy of Classic Bluetooth is lower than that of BLE positioning technology (see Tables 1 and 2). Can the authors provide a link to the work or give their comments?

2) The use of four beacons located at a short distance from the UAS gives a fairly high error in detecting its location. Based on the fact that you use low-energy beacons, you should expect an increase in the target detection error with an increase in the area of the room and the invariance of the number of BLE beacons used. Have authors made any estimates on the number of beacons per unit area needed to determine the location of a target with an error of 1 m (for example)?

Author Response

Reviewer #2:

The methodology of determining UASs' position using low-energy Bluetooth beacons is presented in the paper. The article is well-structured. The experimental data for 2D and 3D position systems are presented (but in static regime only). The references cited in this paper appropriate and relevant to this research.  Some comments:

1) It is not entirely clear why the accuracy of Classic Bluetooth is lower than that of BLE positioning technology (see Tables 1 and 2). Can the authors provide a link to the work or give their comments?

In Table 1, an overview of various indoor positioning technologies is presented, as referenced in [29], we have included a comparison with the results of our proposed method. Table 2 further highlights key distinctions between BLE and Classic Bluetooth, with a particular focus on energy consumption, communication protocols, packet size, and data throughput. The primary differences include BLE’s enhanced energy efficiency, though at the cost of reduced bandwidth and smaller packet sizes compared to Classic Bluetooth, as detailed in [30], [31], and [32].

Additionally, Classic Bluetooth requires over 100 ms to wake from sleep mode (Sleep Time), transmit data, and return to sleep mode, whereas BLE accomplishes this sequence approximately 15 times faster. This reduced scanning time in BLE conserves battery life, particularly beneficial when broadcasting smaller data packets, thereby improving data accuracy. To substantiate this, an additional reference ([33]) has been incorporated into the manuscript.

2) The use of four beacons located at a short distance from the UAS gives a fairly high error in detecting its location. Based on the fact that you use low-energy beacons, you should expect an increase in the target detection error with an increase in the area of the room and the invariance of the number of BLE beacons used. Have authors made any estimates on the number of beacons per unit area needed to determine the location of a target with an error of 1 m (for example)?

Your observation is valid. However, in these initial tests, our approach involved establishing a designated area by positioning the beacons within close proximity to one another, specifically, at distances less than their maximum transmission range of approximately 10 meters. This configuration was designed to minimize potential issues such as signal attenuation, multipath effects, reflections, and shadowing, which could otherwise result in signal loss. Within this controlled environment, our method achieved an accuracy of less than 1 meter (specifically, between 0.40 and 0.60 meters). Based on these promising results, our future objective is to expand coverage to larger areas by implementing a “mosaic” strategy. This would entail covering extensive areas by creating multiple smaller, controlled zones similar to the test area used in this study, ensuring consistent performance across a broader field. This approach has been included into manuscript, in the Conclusions Section (Lines 495 - 499).

This manuscript is a resubmission of an earlier submission. The following is a list of the peer review reports and author responses from that submission.

Round 1

Reviewer 1 Report

Comments and Suggestions for Authors

1.       This paper proposed an RSSI-based BLE localization system for UAS. The main content and idea are the same as the published conference paper Ariante, G., Ponte, S., & Del Core, G. (2022). “Bluetooth 11 Low Energy based Technology for Small UAS Indoor Positioning”. Proc. 2022 IEEE 9th International Workshop 12 on Metrology for AeroSpace (MetroAeroSpace) (pp. 113-118). IEEE. The only difference in method is the new paper includes a DDC process. However, it is unclear how to mitigate the DDC for GNSS localization to the RSS localization. There is no formula description in Section 2.4 but just some qualitative description. I think it is not sufficient for a research paper. Moreover, the extension of this paper is limited compared to the previous conference paper. I can hardly approve that the incremental quantity is enough.

2.       On the other hand, the proposed method is common, including estimating the ranges from RSS, trilateration and Kalman filtering. There are a lot of more sophisticated algorithms localizing the source from RSS measurements straightforwardly and do not have to estimate the distance first. The novelty is weak, and the contribution is not sufficient.

3.       The localization problem (19) is a quadratic equation, both x and y should have two roots, why choose the roots like (20)-(23)?

4.       The simulation configuration is unclear. Why is the noise in RSSI bounded in Fig. 10(a)? It seems the noise in RSSI is discrete, is it reasonable?

5.       “where n was obtained by 1D interpolation of the RSSI values” What is the 1D interpolation you mentioned here?

6.       The performance evaluation is not sufficient by just showing one result in Fig. 17. More statistics, such as RMSE, mean error and bias, should be considered.

7.       Where is the reference for DDC?

Reviewer 2 Report

Comments and Suggestions for Authors

1.Experimental setup area is too small for a actual environment.

Reviewer 3 Report

Comments and Suggestions for Authors

The paper addresses the challenging problem of UAS localization in indoor environments, particularly in GNSS-denied scenarios. The motivation for using Bluetooth Low Energy (BLE) beacons and the application of the Differential Distance Correction (DDC) technique is well-founded. The utilization of the Differential Distance Correction (DDC) technique, inspired by techniques used in GNSS (DGNSS and RTK), is innovative and provides a potential solution to enhance the accuracy of position estimation in the presence of multipath and signal variability.

The paper provides a detailed methodology, explaining the mathematical model, the setup involving Raspberry Pi and Arduino Nano 33 BLE, and the trilateration process. This clarity aids in understanding the proposed approach. The inclusion of experimental results and system performance analysis adds credibility to the proposed methodology. Demonstrating the feasibility and reduction of position uncertainty through practical experiments strengthens the paper.

Here are some suggestions for possible improvements:

While the paper mentions experimental results and the reduction of position uncertainty, a more detailed discussion on these results would be beneficial. Include performance metrics, comparisons, and discussions on scenarios where the proposed approach excels or faces challenges. Besides, it is suggested to introduce more relevant work on the application of Raspberry Pi 4B for laboratory tests, such as: https://doi.org/10.1007/s00170-022-10335-8

Discuss potential limitations and challenges of the proposed method. This could include factors that might affect the performance in real-world applications, such as interference, scalability, or the impact of environmental conditions. Provide a brief comparison of the proposed BLE-based IPS with existing indoor positioning methods.

Include details on the practical implementation challenges, if any, encountered during the setup involving Raspberry Pi and Arduino Nano 33 BLE. This would provide insights for researchers or practitioners looking to implement a similar system.

The paper presents an innovative approach to UAS localization in indoor scenarios using BLE beacons and the Differential Distance Correction (DDC) technique. The detailed methodology and experimental validation contribute to the overall strength of the paper. Addressing the suggestions for improvement will enhance the clarity and completeness of the manuscript.

Comments on the Quality of English Language

The quality of English in the manuscript is generally good, with clear and coherent language.